# A Treatment Decision Support Model for Laryngeal Cancer Based on Bayesian Networks

**DOI:** 10.3390/biomedicines11010110

**Published:** 2023-01-01

**Authors:** Aisha Hikal, Jan Gaebel, Thomas Neumuth, Andreas Dietz, Matthaeus Stoehr

**Affiliations:** 1Head and Neck Surgery, Department of Otorhinolaryngology, University Hospital Leipzig, 04103 Leipzig, Germany; 2Innovation Center Computer Assisted Surgery (ICCAS), Faculty of Medicine, University Leipzig, 04103 Leipzig, Germany

**Keywords:** therapy decision support system (TDSS), Bayesian networks (BN), tumor board, laryngeal carcinoma

## Abstract

The increase in diagnostic and therapeutic procedures in the treatment of oncological diseases, as well as the limited capacity of experts to provide information, necessitates the development of therapy decision support systems (TDSS). We have developed a treatment decision model that integrates available patient information as well as tumor characteristics. They are assessed according to their relevance in evaluating the optimal therapy option. Our treatment model is based on Bayesian networks (BN) which integrate patient-specific data with expert-based implemented causalities to suggest the optimal therapy option and therefore potentially support the decision-making process for treatment of laryngeal carcinoma. To test the reliability of our model, we compared the calculations of our model with the documented therapy from our data set, which contained information on 97 patients with laryngeal carcinoma. Information on 92 patients was used in our analyses and the model suggested the correct treatment in 419 out of 460 treatment modalities (accuracy of 91%). However, unequally distributed clinical data in the test sets revealed weak spots in the model that require revision for future utilization.

## 1. Introduction

Decision-making in oncology requires considering a large amount of heterogeneous information. Experts must evaluate the current disease as well as secondary diseases in addition to personal and social background. In some cases, not all the relevant information can be considered appropriately. This may be due to a lack of available information, the limited memory of the experts, or the shortage of time in everyday clinical practice. Treatment decisions in oncology are generally made in specialized tumor boards. In these meetings, experts from various specialties contribute to the decision-making process to find the optimal treatment tailored to each patient. In some cases, important information is underestimated or disregarded due to the above-mentioned problems of the tumor board. Therefore, many experts advocate treatment support systems [1]. These can help experts to make treatment decisions in everyday clinical practice by suggesting the optimal therapy. These support systems should serve as additional help for the experts and should certainly not replace them.

Treatment decision support systems (TDSS) help to process large amounts of data and to determine the influence of the various pieces of information.

A growing proportion of decision support systems are based on the application of machine learning techniques such as neural networks or deep learning. In this process, systems learn to recognize patterns in data that were not known before or classify information based on found causalities (e.g., for diagnostic or therapeutic decisions) [2,3]. However, these approaches require large amounts of structured data to train and validate the models. Usually, and specifically in complex clinical settings like oncology, these training data are not available.

Another method to develop a decision model is the manual, expert-based approach. Bayesian networks (BN) are suitable structures for manually creating decision models [4].

In a BN, a variety of information entities can be combined to provide flexible and transparent decision representation as well as mathematically accurate and reproducible inferencing (e.g., therapy decision, outcome, comorbidities, or quality of life) [5]. A BN is a probabilistic graphical model that represents a network of categorial variables and their conditional probability distributions. In direct causal dependencies, two directly dependent variables are connected by a directed edge; from a parent node to its child node [5]. Variables describe clinical states such as diseases (e.g., tumor size, location, infiltration of certain structures), symptoms, and complications. For each variable, its states can be either binary (true or false), or if necessary, the probability distribution between 0 and 1 to represent the occurrence distributed among them. Graphical models reflect the causal structure of the variables. The conditional probability tables (CPTs) reflect causal influence from medical evidence [6,7].

Models already exist that function as decision support in clinical practice. BN have been used for decision support in the treatment of lung cancer by Sesen et al. [8]. Leibovici et al. created a model to assist in selecting the most appropriate antibiotics. This model reduced the use of broad-spectrum antibiotics and, consequently, the hospital stay in patients [9]. This suggested the proper antibiotic treatment in up to 85% of 1203 patients in their evaluation. There are other works that created models as therapy decision support but have not yet been applied in practice. Huehn et al. used Bayesian networks for decision support in immune checkpoint blockade in recurrent/metastatic (R/M) squamous cell carcinoma of the head and neck (HNSCC). In their work, the model gave correct probabilities in 84% of all treatment decisions in 25 R/M HNSCC cases tested [10]. Cypko and Stoehr created a model for laryngeal carcinoma and validated a subnetwork that focused on TNM staging [11].

Since laryngeal carcinoma is a common disease in otorhinolaryngology and takes up a large share of the tumor board discussions, we explored the approach and process of model development for treatment decisions for this disease. Laryngeal carcinoma is the third most common malignant tumor in the head and neck region. In Germany, the relation between men and women is approximately 7:1 [12,13]. The main risk factors for laryngeal carcinoma are chronic tobacco and alcohol abuse, especially in combination [14,15,16]. Other risk factors include occupational causes such as exposure to asbestos, ionizing radiation, or exposure to coal products and tar products [17]. Diagnosing laryngeal carcinoma requires a clinical examination. This includes clinical inspection, imaging, panendoscopy, and stroboscopy (= assessment of the vibration behavior of the vocal folds) [18,19,20]. The prognosis of laryngeal cancer depends mainly on the localization, the staging, and, in the case of primary surgery, the status of the resection margins (R status). The larynx is divided into supraglottic, glottic, and subglottic regions. The TNM classification is internationally accepted for the staging of head and neck malignancies. The T stage describes the extension of the primary tumor, the N stage describes the affection of the regional lymph nodes, and the M stage describes distant metastases. Furthermore, the preepiglottic space is of particular importance for tumor extension [21].

Currently, the most effective and widely accepted methods for the treatment of laryngeal carcinoma are surgical therapy, radiotherapy, radiochemotherapy, and immunotherapy. Guidelines for therapy of laryngeal carcinoma are, for example, the NCCN guidelines or the German S3 guidelines. Surgical treatment of laryngeal cancer comprises total or partial laryngectomy, transoral robotic, or laser-assisted surgery. In transoral resection of T1/T2 glottic carcinomas, a safety margin of at least 1 mm should be maintained, whereas a safety margin of at least 5 mm is mandatory for T3 and T4a carcinomas. Radiotherapy can be used alone or in combination with systemic chemotherapy (chemoradiation) [22,23]. For extended or incurable tumors, chemotherapy is necessary. Treatment with chemotherapeutics involves the use of cytotoxic agents that block the growth of malignant tumor cells by various mechanisms [24]. Depending on the tumor status and the patient’s general condition, the type of systemic therapy is chosen. For patients in good general condition with locally advanced disease, concurrent cisplatin and radiation therapy are recommended [23]. Advanced laryngeal cancer can also be treated with immunotherapy, e.g., epidermal growth factor receptor blocking (cetuximab) [24], or checkpoint inhibitors, such as nivolumab or pembrolizumab [23].

We extended the previous work of Cypko et al. [6] and Stoehr et al. [25] by extending their approach for laryngeal cancer support as well as integrating adequate therapy options with a real-life validation effort.

In this work, we describe the procedure and assessment of an expert-based development of a BN model for treatment decision support of laryngeal cancer cases in the head and neck tumor board. In the validation of our model, specific patient information was implemented into the model. In this way, the therapy options recommended by the model were compared with the actual treatment decision in the tumor board.

The work at hand contributes to the field of applied medical informatics by:Presenting a user-centered and expert-based approach to decision modeling;Formally expressing the approach to a decision pathway for laryngeal cancer treatment options;Validating the results using real patient data;Discussing strengths and weaknesses in approaching expert-based decision support.

## 2. Materials and Methods

To create a tumor decision model, it was necessary to consider all aspects that are important for deciding the optimal therapy method. The weighting of individual pieces of information had to be considered, priorities had to be set, and direct and indirect causalities of conditions and therapy methods had to be taken into account.

All diagnostic as well as general information entities of the patient, specifically regarding the tumor were collected. Tumor type, tumor size, tumor spread or infiltration, lymphogenic as well as hematogenic metastasis, etc. were elicited, and the TNM classification was used to determine the tumor stage.

We base our work on the one of Cypko et al., where a TNM staging subnetwork was developed with information on the tumor physical extension (according to TNM staging), comorbidities, genetic and molecular factors, therapy options, risk factors, complications, and quality of life. The model covered the variables relevant to the tumor board, and their causal and probabilistic relationships [6]. The prior results of the TNM model represent the basis of our therapy model.

In our work, reports of tumor board procedures and decision-making strategies, as well as recommended, planned, and actual therapies for laryngeal carcinoma, have been compiled. In patients with laryngeal cancer treated at University Hospital Leipzig from 2017 to 2020, all information entities important for therapy decisions were collected and analyzed.

Based on the information collected and the treatment decisions made in the tumor board of the University Hospital Leipzig, the therapy decision model should function as an aid in the tumor board and be able to recommend the optimal therapy method individually for each patient, taking into account all their findings and information. In this work, the focus was on staging, so treatment options were inferred from TNM staging.

### 2.1. Medical Preliminary Consideration

The tumor state with the different T, N, and M stages as well as the different therapy options and the prerequisites for the implementation of the possible therapies are explained in the following paragraphs.

We created a map that represented the patient information as well as their correlations and causal dependencies. We established arches between entities describing the patient’s condition and the resulting decisions and actions that result from their possible manifestation. The current TNM classification of laryngeal carcinoma works as a source in describing tumor state [21]. For decisions involving surgical therapy, the NCCN Guidelines 2021 were used as the source [23]. This map served as the basis for the technical realization as a BN.

We focused primarily on the TNM stage, as this information was most crucial in the tumor board when making treatment decisions. In addition, since there was not enough data within the dataset to consider detailed information on every patient, it could not be tested and therefore was not modeled. After reviewing the model with ENT experts and assessing the clinical data basis, we reduced the map. The reduction was necessary to allow us a technical realization and validation but also needed to be still clinically sound. In further reviews with ENT clinicians, we secured the clinical significance but also downsized the desired model.

### 2.2. Creating the Model

We utilized the software GeNIe Version 2.2, distributed by Bayesfusion (https://www.bayesfusion.com/genie/), for model development and validation [26]. Information relevant to the treatment decision of laryngeal cancer is specified in nodes. In these nodes, the different characteristics or states matching treatment decisions are specified as states. For example, a node for the “T-stage” was created representing the different T-stages, which are classified according to the TNM classification. This is illustrated in Figure 1.

For the T category, a node was created with subcategories T1-T4b as well as TX, T0, and TIS. For metastases to the regional lymph nodes, a node was generated and the possible N stages were given as the states of this event. The same was performed for the distant metastases, which were listed in an M node. Since the T, N, and M categories are causally related to all therapeutic options, this was taken into account in the graph as well as in the evaluation.

In the next step, a therapy sub-model was created. In this sub-model common as well as all overall possible therapy methods were listed. For the possible therapy options, the exact type or description of the respective therapy was presented as subitems.

The prerequisites for specific therapies to be performed were elicited and placed above the possible therapeutic options in the model. Of these prerequisites, edges were placed in the direction of the therapy nodes. Thus, a therapy could only be selected if the prerequisite above it was fulfilled.

When creating the nodes, we distinguished on the “logical level” between observable nodes and calculated/inferred nodes. Observable states include patient information that we insert into the model. Our target variables, the therapy options, on the other hand, can occur in a distributed manner and express probabilities of success and are calculated/inferred nodes. For example, in a given constellation, surgical therapy may be 70% true and radiotherapy 30% true [27].

### 2.3. Probability Integration

GeNIe is able to infer the probability for certain target nodes. Probability conditions must be entered into the model to represent the causal conditions between nodes. Conditional probability tables in GeNIe contain the individual causal conditions of all combinations of states between two nodes. Using the NCCN guidelines, we entered the specific probability of one state occurring dependent on the parent’s states’ occurrence. The probabilities were entered into the model for which therapy would be selected according to the guidelines for the most varied constellations. In the program described below, the most varied situations with the different stages and patient conditions were specified and the appropriate therapies were selected based on the guidelines.

For example, surgical therapy would have a probability of 74% of being considered for a patient with stage T2, N2a, and M0 laryngeal carcinoma and who is tolerant to chemotherapy. In contrast, a patient with stage T2, N2a, and M1 laryngeal carcinoma would not receive surgical therapy with a probability of 17%. Figure 2 shows an example of a table with the entered probabilities in the different states.

### 2.4. Description of the Data Set

The data analysis included information on 97 patients with laryngeal cancer, who were presented to the head and neck tumor board and treated at the Department of Otolaryngology at the University Hospital Leipzig. Patients with the same health conditions or the same tumor stages were grouped. The analysis included data from 83 men (85.6%) and 14 women (14.4%). Most patients (n = 32) from the data analysis were between the ages of 61 and 70 years (33.0%) followed by the age group between 71 and 80 years with 21 patients (21.6%). From the dataset, five patient cases were excluded from the analysis because they contained inconsistent data from the health records. Thus, 92 patient cases were analyzed. Table 1 shows the frequencies in detail. 

Most of the tumors were in advanced stages T3/T4a, as shown in Table 2. Surgical therapy was the most common treatment modality among patients from the data set. In some patients, primary tumor extension or metastases did not allow for surgical intervention. In addition to the poor general condition and therefore insufficient anesthetic ability, the missing complete tumor resectability with adequate safety margins prevented surgical therapy in 11 patients (11.3%). An insufficient anesthetic ability was present in two cases, where patients had very poor general condition. Surgery was a performed therapeutic option in the data set with 80 patients (82.5%). Chemoradiation ranked second, which was performed in nine patients (9.3%), whereas radiotherapy alone was performed in three patients (3.1%).

### 2.5. Validation Process

For validation, a set of information on 92 patients who were treated at the University Hospital Leipzig due to a laryngeal carcinoma was available to us. We entered the data into the GeNIe as a CSV file and used its internal validation algorithm to test the model. We configured the five treatment nodes to be the target for the validation, meaning GeNIe inferred them given the data from each patient case and compared it to the given data from each treatment option. The outcome of the validation test will be described in the Section 3.

We also utilized GeNIe’s internal training feature to test the outcome of training the model’s conditional probabilities using real-life data. We used a 10-fold cross-validation with uniform parameter initialization to train and test. However, given that we only had 92 test cases, the cross validation performed worse and the resulting accuracy was lower. This confirmed our efforts in creating a decision model manually with the precondition of sparse clinical data sets.

## 3. Results

### 3.1. The Treatment Model for Laryngeal Carcinoma

In Figure 3, we illustrate the therapy model including the different descriptions of the tumor as well as the therapy methods. The model consists of nine nodes. The orange-colored boxes represent the T, N, and M stages of the laryngeal tumor. Based on the TNM classification, the T stages range from TX-T4b, the N stages range from NX-N3b, and the M stages range from MX-M0 [21]. The pink-colored boxes reflect the different therapeutic modalities of laryngeal carcinoma. The yellow-colored box describes the tolerability for chemotherapy, which is a prerequisite for the implementation of chemotherapy.

Our model consists of variables describing the tumor based on T-, N- and M-stages as well as variables related to prerequisites for certain therapies and the different therapy options. The model consists of nine nodes and 18 edges. Figure 4 shows our model divided into the categories of TNM-stages, therapy options and prerequisites for certain therapies.

### 3.2. Validation of the Treatment Model

Our analysis showed an overall accuracy of 91% for all five target nodes regarding the model calculation in comparison with the documented tumor board decision. This is the result of multiplying the number of patients in our data set (n = 92) and the possible therapy options, a total number of 460 treatment option combinations. Out of these treatment options, 419 were chosen correctly by our model (91%). For the decision regarding chemotherapy, the accuracy of the model was 73%. In many cases, the model showed deviating results compared to the recorded tumor board decision. This may be due to the smaller number of cases compared to surgical therapy leading to greater imprecision. Another reason can be that patients often have a say in the choice of therapy and opt for a treatment method that is not the first choice according to the guidelines. Regarding radiochemotherapy, the model calculated the treatment with an accuracy of 93%. For surgical treatment, the model achieved a correct recommendation in 91% of the cases. The accuracy for radiotherapy alone was 98%. For immunotherapy, the accuracy reached 100%. Since this therapy is currently being established and therefore no data were available in our data set, this result is not significant.

Table 3 presents the accuracy of the model calculation with respect to the different treatment modalities. Figure 5a–c shows the receiver operating characteristic (ROC) curves for the three main treatment nodes: surgery, radiochemotherapy, and radiotherapy, respectively. The other two treatment options were omitted because there were no positives in our data. Generally speaking, ROC curves show the relation of sensitivity and specificity of a classifier. A good performance of a classifier would be indicated by a curve that moves close to the top-left corner. The ROC curves demonstrate high predictive performance of the model for the majority classes and acceptable performance for rare therapy decisions (see Table 4 for class frequencies). However, one can clearly see the difference between the curves regarding one therapy option. One option is better defined with high validity than the other (e.g., predicting the performance of surgery rather than against it). This can be looped back to the evidence base we used. There, most patients with similar staging received similar therapies. Hence, the prediction tends to be the evidentially more suitable treatment. This can also be seen in the area under the curve (AUC), which in all three cases shows the tendency to favor one treatment option over the other.

Figure 6a–c displays the precision-recall curves for the three therapy options: surgery, radiotherapy, and radiochemotherapy. These curves plot the positive predictive value against the true positive rate. Curves for chemo- and immunotherapy were omitted because of missing positive cases in the test data, and therefore graphs would be unusable. All curves show a high sensitivity for one specific option. Figure 6a shows high sensitivity toward larynx surgery (blue curve), Figure 6b,c shows high sensitivity against the performance of either radio- or radiochemotherapy (orange curves). This means that with a high recall there are few to no false positive predictions. However, the opposite therapy options (orange curve in Figure 6a, blue curves in Figure 6b,c, respectively) reach high recall only with lower precision. Our model’s prediction against surgery or for radiochemotherapy, respectively, reaches a precision of roughly 50% with a recall of only 50%. The precision of outcome values for radiotherapy does not exceed 20% precision (see Figure 6c).

We calculated the F1 score as a harmonic mean of precision and recall. Table 4 shows the results of our model. It is also clearly visible how either one therapy option tends to favor only one of its possibilities. Both precision-recall curve and F1 display the flaws in the test data, which are unbalanced in terms of containing all possible values in a suitable distribution. This implies that statements from our model should be individually assessed in regard to the model’s sensitivity.

## 4. Discussion

The aim of our work is to create the approach of a decision model for laryngeal cancer as a potential basis for a TDSS for the head and neck tumor board. To find an optimal patient-specific treatment option, the processing of large amounts of data from many different sources is required. This process can be supported by TDSS. Especially by multidisciplinary teams, for example in tumor boards, such systems are increasingly applied [28]. TDSS based on BN can play a significant role because they provide flexible and transparent decision modeling, as well as mathematically accurate and reproducible recommendations [4,5].

In a validation study of 92 patient cases discussed in the head and neck tumor board of the University Leipzig, the model showed good results in calculating the optimal treatment from primary patient data compared to the actual tumor board recommendation.

For treatment options that were recommended less frequently, however, our model showed deviating results. Here, our assessment of the model performance is inconclusive due to very few cases for validation. Poor validation metrics could indeed indicate poor predictive accuracy or originate from statistical noise. So far, the best patient-specific treatment is based on the most frequently used therapy, as this is evidence-based. This does not necessarily reflect the success of the therapy, but only the frequency or probability of use.

Since immunotherapy is currently being established, there is insufficient data for patients treated with immunotherapy in our dataset, which contains retrospective patient data from previous years. Because of this, the validation of the immunotherapy node was not possible with clinical data. However, the node was still integrated into the model, as cases will occur more frequently in laryngeal cancer treatment.

Previous models in head and neck oncology, such as Huehn et al. already showed acceptable results for clinical models in a molecular setting [10]. In addition, it was stated that abstracted patient factors and generalized medical guidelines must be adjusted to the individual case and verified by real-world data, despite an accurate processing model with appropriate results. Huehn et al. also indicated that their model represents a tradeoff between the complexity of the model and the ability to evaluate each possible individual decision with reasonable certainty [10]. This was also a pitfall in our approach to modeling. We needed to find a realistic, medical sound depiction, yet rely on a limited data set of test cases. However, combining different views and previous works, e.g., [23] and [6], promises to overcome individual shortcomings.

A treatment decision model for oropharyngeal carcinoma was designed by Buyer et al. [29]. In this work, it was discussed that the determination of the metrics presented was highly dependent on the underlying data set. Thus, they also assumed that the results were limited and could be expanded with the integration of more or different other data sets. Buyer et al. suggested that the approach presented could likely adapt to real-world causal effects such as a reduction in the need for adjuvant treatment when R0 resection with clear margins was achieved. They also wrote that the resulting sets of their work were purely categorical, and this would have allowed their approach to be compared with other methodological solutions that are also able to account for state differences between variables such as the Goodall measure or likelihood-based methods [29].

In our analysis, it was noticeable that uncertain therapy methods were suggested for information that was not clearly defined, such as Tx (= primary tumor cannot be assessed), because no correct probability can be generated if the information is unclear. In our model, we restricted ourselves to the TNM nodes as well as the nodes for the possible therapies. Originally, the model was much more complex. However, this was significantly reduced because we would not have been able to validate the causalities since we did not have extensive enough test cases. Not being able to demonstrate that certain conditions really influence decisions is a necessity in clinical decision support. In addition, an excessive number of nodes, and thus information, can lead to limitations in the computability of the model. Huehn et al. also described that an immense increase in the size of the model can complicate the process of determining the individual probabilities of individual connections between two nodes [10].

Cypko and Stoehr presented a model for laryngeal carcinoma and validated a subnetwork that focused on TNM staging [11]. It has to be noted that the TNM classification is a clearly defined system, whereas treatment decisions are much more complex and therefore diverse, as shown in the different options for similar circumstances. In this respect, the presented treatment showed a reasonable overall accuracy of 91%.

While the preliminary work of Cypko and Stoehr focused on TNM staging, in our model, reference was made exclusively to preoperative therapy [11]. This means that our model is suitable for the treatment of primary laryngeal cancer patients. For adjuvant treatment as well as for recurrences, the creation of separate models is planned and not the focus of this study.

In our work, we focused primarily on the TNM classification as a relevant factor. The validation study of the model with 92 patient cases may have limited validity. However, this model serves as an advanced concept for the use of treatment decision support systems in the tumor board and an extension of the model is planned. As new therapeutic modalities such as the various chemotherapeutic agents as well as immunotherapies are increasingly used, an extension of the model or its integration with other models, e.g., Huehn et al. and Cypko and Stoehr is necessary for the future [10,11]. Although these therapy methods were not used in our dataset they were nevertheless integrated into the model because they are relevant for patients in palliative situations. In the time we treated, there were no matched patients included in the analysis. In addition, more patient data will be available for validation studies as newer therapies increase in use. Despite the limited amount of patient cases, the results of our study are promising.

The model expansion also aims to use a larger data set as well as a larger set of parameters. Similarly, collecting data from different centers in different regions would be beneficial. Additionally, the method needs to be validated in different backgrounds to prove its potential.

## 5. Conclusions

A therapy decision support model serves as an aid for the experts, for example in the tumor board. It can be used as a small feature within a clinical information system in preparation for the tumor board to illuminate possibilities of the therapy approach in advance or it can be used in the tumor board itself. However, it should not be used as a basis for decision-making. The experience and knowledge of the experts should still be taken into account. It must always be considered in the context of clinical practice.

Our decision model for laryngeal cancer was created by integrating the most relevant parameters for a general treatment decision. The BN model was validated in a retrospective analysis of 92 patients who were presented on the tumor board of the University Hospital Leipzig from 2017 to 2020. In this analysis, the calculation of the model from primary data taken from the patient record of 92 patients was compared to the treatment decision of the tumor board. Because of conflicting data, information from five patient cases was excluded from our analysis so as not to bias the results. The analyses of our model showed an overall accuracy of 91.0%.

Therapy decision support systems can be a real asset in everyday clinical practice. For the development of a decision model, a large balanced data set is indispensable to achieve better results. To prove that the decision model can be used in practice, validation with patient data from clinical practice has to be performed. With a rather unbalanced data set, accurate predictions cannot be made for certain aspects, in our case chemotherapy and immunotherapy. With a larger and more homogeneous data set, better results can be obtained.

Overall, however, the structure of the model and the formalism are very important. Evidence-based causalities were set. The probabilities that were entered into the GeNIe model are based on guidelines that are also used in clinical practice. Extension and maintenance of the model are important and possible at any time. As soon as new knowledge or new findings occur, the model can be adapted to them.

These results will be tested prospectively, and we hope that further optimization and validation will lead to a beneficial clinical decision support system that provides transparent and comprehensive assistance in the decision-making process.

## Figures and Tables

**Figure 1 biomedicines-11-00110-f001:**
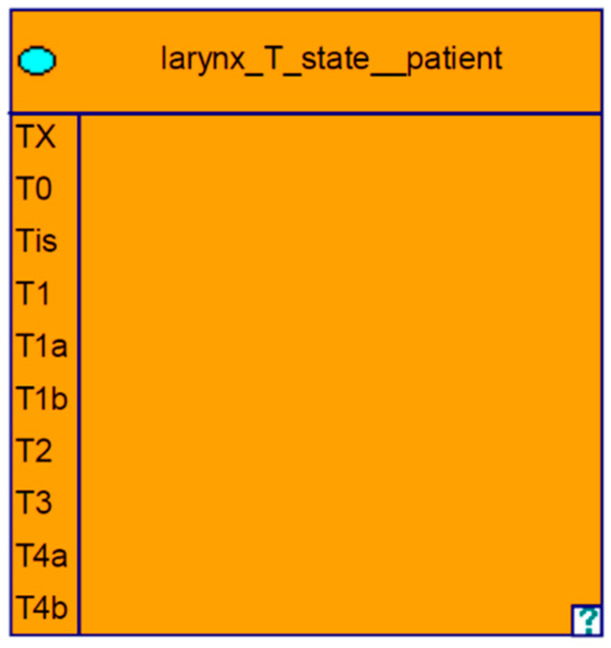
Node: T category with the different states.

**Figure 2 biomedicines-11-00110-f002:**
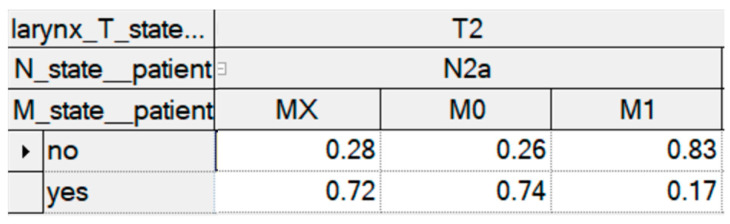
Probabilities of surgical therapy in the different TNM constellations.

**Figure 3 biomedicines-11-00110-f003:**
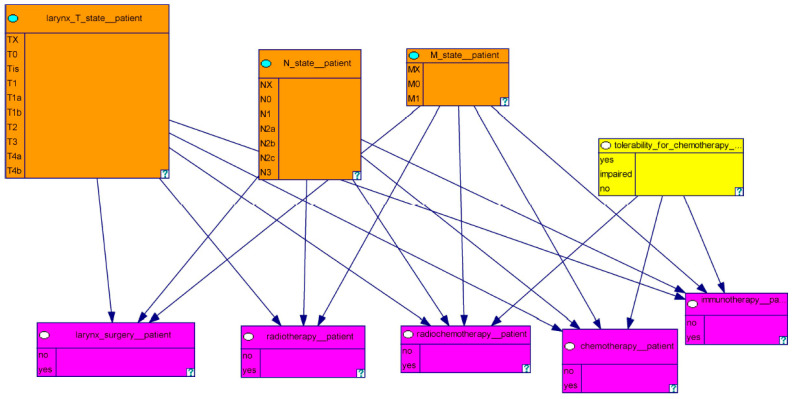
This figure shows the therapy model with the different descriptions of the tumor and the therapy methods.

**Figure 4 biomedicines-11-00110-f004:**
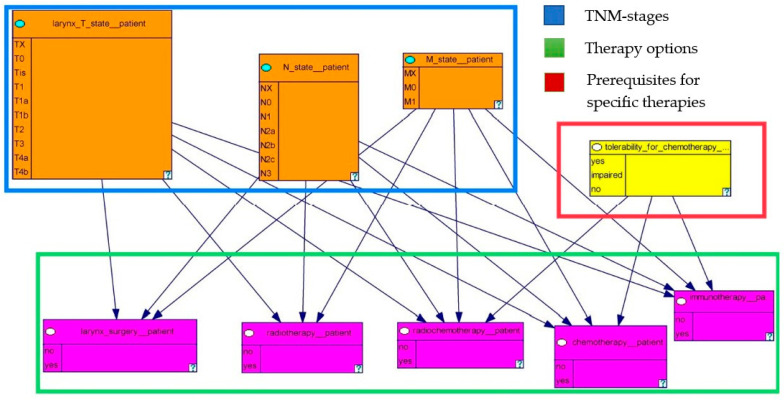
The therapy decision model grouped into the different categories.

**Figure 5 biomedicines-11-00110-f005:**
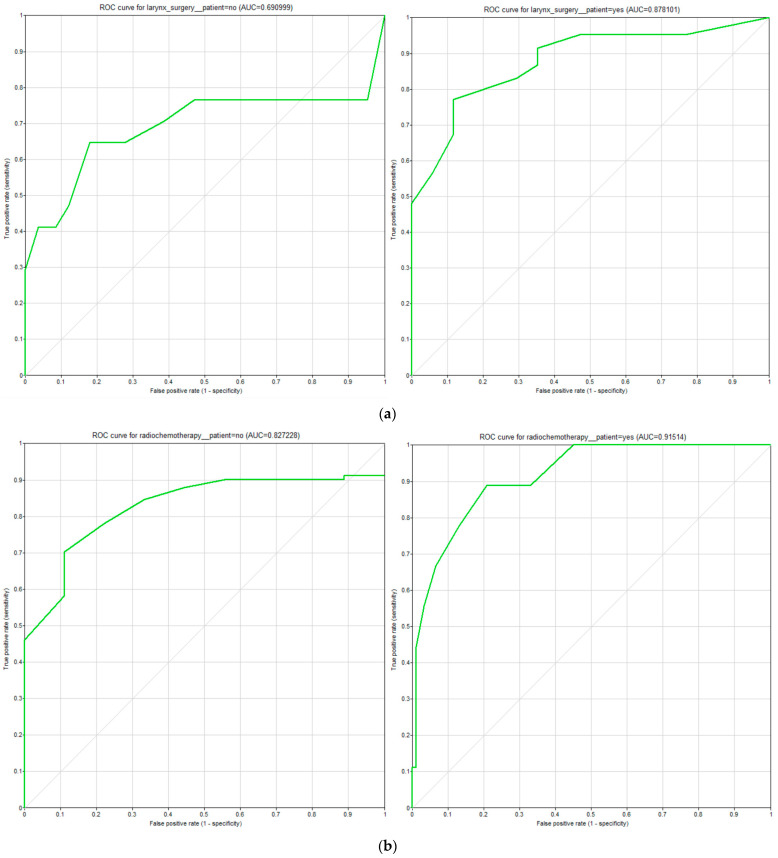
(**a**) ROC curves for treatment option “surgery”, (**b**) ROC curves for treatment option “radiochemotherapy”, (**c**) ROC curves for treatment option “radiotherapy”.

**Figure 6 biomedicines-11-00110-f006:**
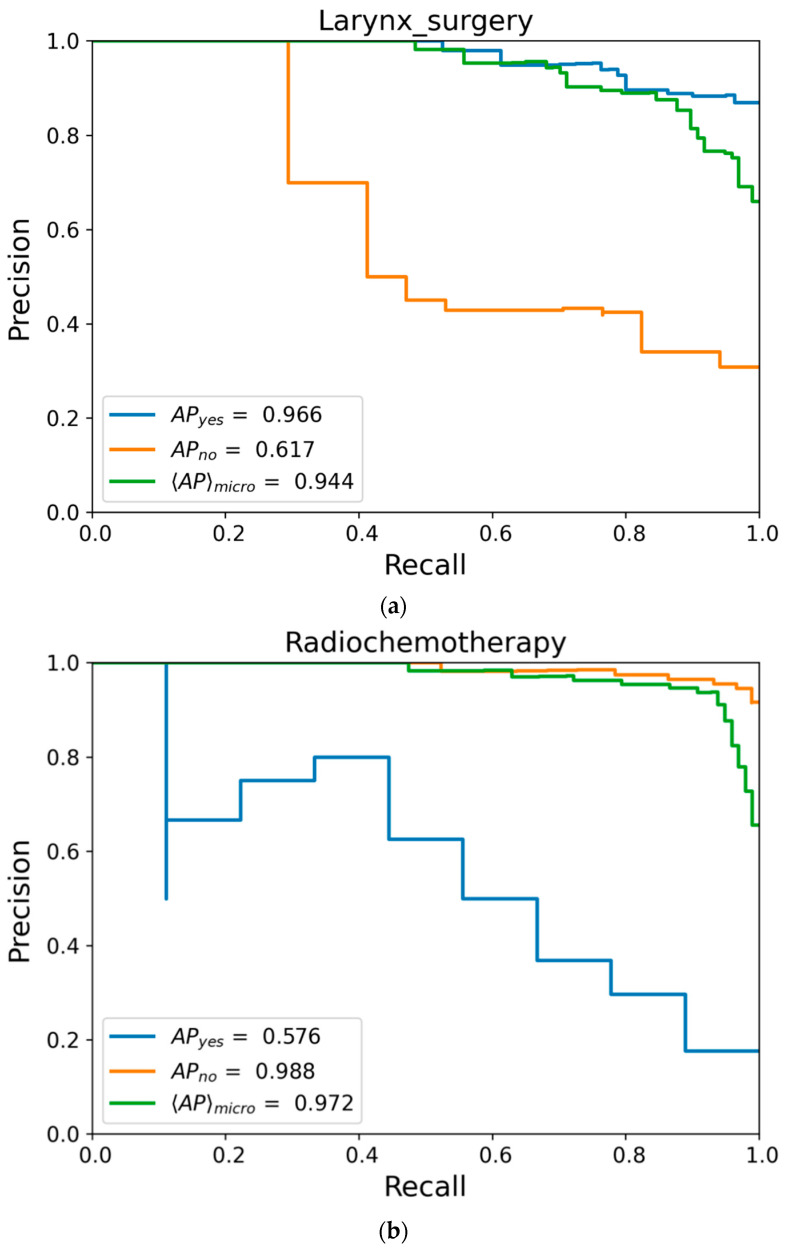
(**a**) Precision-recall curve for surgery; (**b**) Precision-recall curve for radiochemotherapy; (**c**) Precision-recall curve for radiotherapy; *AP* = average precision (area under precision-recall curve), *<AP>_micro_* = micro-averaged over both classes.

**Table 1 biomedicines-11-00110-t001:** Statistical summary of the patient-related features for therapy decision.

Patient-Related Features n = 97		Absolute Frequency	Relative Frequency
Gender	Male	83	0.856
	Female	14	0.144
Age group	51–60	30	0.302
	61–70	32	0.330
	71–80	21	0.216
	81–90	12	0.124
	unknown	2	0.021

**Table 2 biomedicines-11-00110-t002:** Statistical description of the diagnosis-related factors for therapy decision.

Diagnosis-Related Features n = 97		Absolute Frequency	Relative Frequency
T State in total	Tx	1	0.010
	Tis	1	0.010
	T1	5	0.052
	T1a	19	0.196
	T1b	8	0.082
	T2	16	0.165
	T3	22	0.227
	T4a	23	0.237
	T4b	2	0.021
T State without surgical intervention	Tx	1	0.010
	T1b	1	0.010
	T3	5	0.052
	T4a	8	0.082
	T4b	2	0.021

**Table 3 biomedicines-11-00110-t003:** Accuracy of the model calculation regarding the different treatment modalities.

Treatment	Accuracy in Absolute Numbers	Accuracy in Relative Numbers
All nodes	419/460	0.91087 (91%)
Chemotherapy	67/92	0.728261 (73%)
Radiochemotherapy	86/92	0.934783 (93%)
Larynx surgery	84/92	0.913043 (91%)
Radiotherapy	90/92	0.978261 (98%)
Immunotherapy	92/92	1 (100%)

**Table 4 biomedicines-11-00110-t004:** F_1_ measure for larynx model.

Treatment	Positive Cases	Negative Cases	F1 of Positive Class	F1 of Negative Class	Weighted Average of F1
Chemotherapy	0	92	0.843	0	0.843
Immunotherapy	0	29	1	0	1
Larynx surgery	80	12	0.556	0.952	0.896
Radiochemotherapy	9	83	0.965	0.571	0.926
Radiotherapy	3	89	0.989	0	0.968

## Data Availability

Not applicable.

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
