# Peer review of "A Treatment Decision Support Model for Laryngeal Cancer Based on Bayesian Networks"

_biomedicines, 2023, doi:10.3390/biomedicines11010110_

Round 1

Reviewer 1 Report

Authors apply a software implementing some Bayessian networks algorithms to Laryngeal Cancer data.

I do not see any real contribution beyond the technical report about the results obtained and the introduction explaining the problem and how it can be modeled as states. They do not show any contribution from the algorithmic point of view (just to use a common library package to run the experiments; there are plenty of options to analyze data involving some dynamics) nor deep analysis about the validation of the results. They show some screen captures about how to introduce the data in the software ? From table 2 it is clear tha dataset is highly unbalanced; what did you do about this? What about the statistical analysis of the results (probably they will show no statistically significance difference since in some cases you have 1 or 2 samples per class so I do not see how you can train anything reliably based on 1 or 2 observations. I undertsand it is not easy to get more data in this kind of problems involving physicians, so authors should be more careful about the conclusions and try more ideas than just an available software.

Author Response

Comment of reviewer:

            English language and style are fine/minor spell check required

Response to the reviewer:

            We conducted a check and correction of spelling and style wherever appropriate.

Comment of reviewer:

Does the introduction provide sufficient background and include all relevant references? Can be improved.

Response to the reviewer:

Thank you for this formal comment. We attempted to include all necessary information in our introduction. The introduction describes the problems or potential inefficiency that exist in the tumor board and in general clinical practice, which is why we decided to develop a treatment decision model for laryngeal carcinoma. Introductory information on the topic of therapy decision support systems is included. In addition, we discuss Bayesian networks and other works that used BN . Furthermore, laryngeal carcinoma and its treatment standards according to the guidelines are explained. Also, comparable work is mentioned, which is extended in our research and described in extension. With this additional information, we hope to give the reader an insight into all important aspects of our work.

Comment of reviewer:

Are all the cited references relevant to the research? Can be improved.

Response to the reviewer:

We thank the reviewer for this feedback. We have again rechecked our bibliography to ensure that all relevant information in our work is also cited and, conversely, that our references are relevant to our work. Our bibliography includes, for instance, references on the background of our work, previous work in the field of therapy decision support systems, the guidelines on which our therapy decisions are based, and other references that we considered relevant to our work.

Comment of reviewer:

Is the research design appropriate? Must be improved.

Response to the reviewer:

We thank you for this evaluation. Our work aims to describe the process of creating a treatment decision support model for laryngeal carcinoma. For this purpose, all information necessary for therapy was collected and compiled at the beginning. Information about the tumor and its spread, information about the patient's health status, and other content necessary for decision-making. Using the software, GeNIe 2.0, the model was presented graphically. Another step was to enter the probabilities for the occurrence of the different therapeutic methods under the different circumstances and tumor stages. To check whether the model was valid, the therapies suggested by the therapy model were compared with the therapies actually performed for a data set we collected. This showed promising results for the three main therapies: surgical therapy, radiochemotherapy, and radiotherapy. For the other two therapies, the results are not valid due to the small number of cases. We are aware that there is a need to expand our concept. However, to this point the analysis serves as a proof-of-concept and there are plans to expand the model in the future and also extend the number of cases. So far, we had only the presented number of 92 cases available for validation of our model.

Comment of reviewer:

Are the methods adequately described? Must be improved.

Response to the reviewer:

Thank you for this important point. We now expanded the “Methods” section and gave a more detailed description of the methods and approach of our work. At the beginning of the section, we mentioned what important clinical information was collected, on which work our research is based, which is mainly Cypko et. al., and from which center and which treatment period the data set originates.

In the section “Medical preliminary consideration”, the exact procedure of data collection was described. Here we added important information on the procedure.

The section "Creating the model" explains to the reader how we chose the information for the different nodes and their states. Then we described the creation of the submodel with the placement of the edges between the different nodes.

In addition, it was explained that in creating the nodes we distinguished at the "logical level" between observable nodes and computed/inferred nodes and we explained this further.

In the following chapter, we explained how we entered the probabilities into the model using the GeNIe software. We also described that our treatment recommendations are based on the NCCN guidelines for head and neck cancers.

Then we described our data set in detail, providing data on the ratio of males to females, the frequency in the different age groups, and the frequency of the different T-, N-, and M-stages. Tables 1 and 2 provide detailed information.

In the section "Validation Process" we described the number of patients whose results were actually evaluated, the procedure of importing the patients of our dataset in form of a CSV file into GeNIe, and how the validation was performed.

We also described that we used GeNIe's internal training function to test the result of training the model's conditional probabilities on real data and that we used 10-fold cross-validation with uniform parameter initialization for training and testing. However, because we had only 92 test cases, the cross-validation performed worse and the resulting accuracy was lower. This confirmed our efforts in manually building a decision model under the condition of limited clinical datasets. Finally, we added the information that we abandoned the idea of statistically analyzing the model since the limited dataset would not promise to deliver reliable outcomes and that we chose to take a broader look at the model and assess its accuracy in general. In the future, however, we plan to test the model prospectively ideally in a controlled clinical trial. Still, this is a separate project and future work.

Comment of reviewer:

Are the results clearly presented? Must be improved.

Response to the reviewer:

We thank you for this argument. In our manuscript, we included screenshots that show the reader a step in working with the software as well as the model view. In addition, we now also included an illustration that divides the different nodes of our therapy model into categories. These images are used to illustrate the approach in our work. Figure 1 shows a node as it is represented in our model. The node is labeled at the top with its title, in this case, laryngeal T-State. The column below lists the different states of a node, in this case, the different T-stages, based on Wittekind's TNM classification. The second figure is intended to provide a small insight into the process of entering the probabilities for the reader. When entering the probabilities, the GeNIe software displayed the different possible stages and combinations of the TNM in connection with the different therapy options in a tabular view. Figure 2 shows states for the T2-stage, N2a-stage, and the three possible expressions of the M-stage, namely MX, M1, and M2. For each of these combinations, there are two possible states for the choice of surgical therapy, which are yes and no. This means whether surgical therapy would be chosen for this. Figure 3 shows the therapy model with the different descriptions of the tumor and the therapy methods. This serves as an overall view of the therapy model and shows immediately which information was considered relevant and therefore generated in nodes. As mentioned above, the new Figure 4 shows a classification of the nodes into the categories TNM-stages, therapy options, and prerequisites for specific therapies.

Comment of reviewer:

Are the conclusions supported by the results? Can be improved.

Response to the reviewer:

We thank you for this opinion. In the “Conclusions” section, we described the set of patients from our dataset that we analyzed with our therapy model. Reasons for the exclusion of 5 patients and thus the reduction of our analyzed data set to 92 patients were outlined. The problem of a few patient cases for selected therapy options was discussed. Now we added content in this regard. Due to the accuracy of more than 90% for therapy options that occurred frequently in our dataset, we are confident that with a larger and balanced dataset that can be generated in the next years, significant results can be achieved with our therapy model. So far, the dataset was limited as outlined above.

Comment of reviewer:

I do not see any real contribution beyond the technical report about the results obtained and the introduction explaining the problem and how it can be modeled as states. 

Response to the reviewer:

We thank you for this argument. As already described earlier comments particularly regarding the methods, we expanded the section "Materials and Methods" and described all the steps of our approach and the problems encountered during modeling and validation. Therefore, we clearly see a substantial contribution to the research in developing decision support models by the presented work, both technically and in terms of content.

Comment of reviewer:

They do not show any contribution from the algorithmic point view (just to use a common library package to run the experiments; there are plenty of options to analyze data involving some dynamics) nor deep analysis about the validation of the results. 

Response to the reviewer:

We thank the reviewer for this remark. We supplemented our methods part, and thus also the section on the validation process. Both the procedure and the results of the validation are described with proper extent. 

Comment of reviewer:

They show some screen captures about how to introduce the data in the software?

Response to the reviewer:

In our manuscript, we included screenshots that show the reader a step in working with the software as well as the model view as a whole. In addition, we now also included an illustration that divides the different nodes of our therapy model into categories. These images are used to illustrate the approach in our work. Figure 1 shows a node as it is represented in our model. The second figure is intended to provide a small insight into the process of entering the probabilities for the reader. When entering the probabilities, the GeNIe software displayed the different possible stages and combinations of the TNM in connection with the different therapy options in a tabular view. Figure 2 shows states for the T2-, N2a., MX-, M1-, and M2-stage. with the probabilities for the choice of surgical therapy. Figure 3 shows the therapy model with the different descriptions of the tumor and the therapy methods. This serves as an overall view of the therapy model and shows immediately which information was considered relevant and therefore generated in nodes. As mentioned above, the new figure 4 shows a classification of the nodes into the categories TNM-stages, therapy options, and prerequisites for specific therapies.

Comment of reviewer:

From table 2 it is clear the dataset is highly unbalanced; what did you do about this?

Response to the reviewer:

Thank you for pointing out this fact. Our data set actually includes mostly patients from one category, which is those who have undergone surgical therapy. This is because until now, surgery has been the predominant treatment for laryngeal carcinoma. However, other therapy methods are increasingly used and this will be seen in the next years and decades. So far, surgical therapy is the most frequently used method, at least in German-speaking countries, because of high experience in this field and excellent results. Still, we are aware of this point and discussed it in our work, particularly that in the future an expansion of the model with a larger data set from different centers and countries is aspired. This may lead to a more balanced data set.

Comment of reviewer:

What about the statistical analysis of the results (probably they will show no statistically significance difference since in some cases you have 1 or 2 samples per class so I do not see how you can train anything reliably based on 1 or 2 observations. I understand it is not easy to get more data in this kind of problems involving physicians, so authors should be more careful about the conclusions and try more ideas than just an available software.

Response to the reviewer:

For therapy options with only 1 or 2 patient cases, we cannot speak of significance. As already noted, this is based on our data set, which does not provide more patient cases for these particular therapies. However, this should change in the future as more different therapies are used and also as the data set is extended over the years to come. Our model is a starting point in this direction, and we do not deny the need of further development. In our manuscript, we also noted that for therapies with only few cases, we do not speak of significance. In contrast, especially for surgical therapy, there was a large number of patient cases and this showed an accordance of more than 90%. Therefore, we are confident that the remaining therapies will show high accordance as well if enough cases are available to test this.

Reviewer 2 Report

In this manuscript, the authors have developed a treatment decision model for laryngeal carcinoma. Diagnostic information is complex, and treatment decisions are tough. Artificial intelligence (AI) is valuable as an analytical tool but challenging to apply to something as complex as a cancer treatment. Bayesian network (BN) is suitable for processing complex information and is actually used for the treatment of lung cancer and the selection of antibiotics. The authors used the BN network for laryngeal cancer, the third most common malignant tumor of the head and neck region, and attempted to create a model that supports treatment decisions. As a result, their analysis showed 91% accuracy, indicating that this method may support the treatment of laryngeal cancer. The following minor points need to be addressed.

1.           Please use larger fonts in figures 4, 5, and 6 to facilitate a better experience for the reviewers and readers.

2.           The authors should be more specific about figures 4, 5, and 6.

Author Response

Comment of reviewer:

  1. Please use larger fonts in figures 4, 5, and 6 to facilitate a better experience for the reviewers and readers.

Response to the reviewer:

We thank the reviewer for this important note. The Figures have now been inserted in higher quality and larger format so that reviewers and readers have better access to the graphs. 

Comment of reviewer:

  1. The authors should be more specific about figures 4, 5, and 6.

Response to the reviewer:

We thank the reviewer for this comment. These figures show the receiver-operator-characteristics (ROC) curves for the three main treatment nodes; surgery, radio-chemotherapy, and radiotherapy, respectively. The other two treatment options, chemotherapy, and immunotherapy were omitted because of poorer reliability based on the lack of test data. Generally speaking, ROC curves show the relation of sensitivity and specificity of a classifier. A good performance of a classifier would be indicated by a curve that moves close to the top-left corner. From the ROC curves, it can be seen that all decisions made with regard to the three main treatment nodes were generally of good diagnostic quality. However, it can clearly be seen a difference between the curves concerning one treatment option. One option is better defined with high validity than the other (e.g., predicting the performance of surgery rather than the consideration). This can be attributed to the evidence base we used. There, most patients with similar staging received similar therapies. Therefore, the prediction tends toward the treatment shown to be more appropriate. This can also be seen in the area under the curve (AUC), which shows a tendency to favor one treatment option over the other in all three cases.

Reviewer 3 Report

Overall, this is an interesting and insightful work for the individualized treatment decision for laryngeal cancer. 

Authors conducted the analysis via machine learning and got a model with high sensitivity and specificity.

The methods and results comprehensively described the analysis they have done.

Some minor revisions are needed:

The Figures need to be updated with high resolution.

Figure 2 needs to be remade to help readers better understand the contents instead of the screenshot.

Author Response

Comment of reviewer:

English language and style are fine/minor spell check required.

Response to the reviewer:

            We conducted a check and correction of spelling and style wherever appropriate.

Comment of reviewer:

Are the results clearly presented? Can be improved.

Response to the reviewer:

We thank the reviewer for this important comment. The results for the significance of the statement of our model were presented for the different treatment options. Because of the imbalance of our dataset, which is based on the fact that certain therapy options are preferred, some therapy options were presented more often than others in our analysis. As shown in table 3, a significance of 91% was presented for the therapy options overall. This includes all the therapy options presented, including those that have only been established or are gradually being used more frequently. For surgical therapy, a significance of 91% was analyzed. This means that the recommended therapy option from our model and the performed therapy in the ENT clinic of the University Hospital of Leipzig correlate to 91%. Since this therapy option is by far the most frequent, a significant statement can be assumed. Since chemotherapy and immunotherapy did not occur frequently in our data set, a correlation in terms of significance regarding the calculation of our model and the performed treatment was omitted. For the remaining therapy options namely radiotherapy and radiochemotherapy, the significance of the statement of our model was also listed in table 3. We are aware that our model is in early stages of the project and will be expanded in the future and tested with an extended and balanced data set.

Comment of reviewer:

  1. The Figures need to be updated with high resolution.

Response to the reviewer:

We thank the reviewer for this important note. The figures have now been updated in higher resolution and a larger format so that reviewers and readers allow for better access to the graphs. 

Comment of reviewer:

  1. Figure 2 needs to be remade to help readers better understand the contents instead of the screenshots.

Response to the reviewer:

We thank the reviewer for this important remark. The figure has been uploaded in better quality and a larger format. This illustration is intended to provide a small insight into the process of entering the probabilities for the reader. When entering the probabilities, the GeNIe software displayed the different possible stages and combinations of the TNM in connection with the different therapy options in a tabular view. Figure 2 shows states for the T2-stage, N2a-stage, and the three possible expressions of the M-stage, namely MX, M1, and M2. For each of these combinations, there are two possible states for the choice of surgical therapy, which are yes and no. This means whether surgical therapy would be chosen for this. combination. For each combination and each state, the probability was entered manually. This was based on the NCCN Guidelines 2021. As can be seen in the figure and described in the paragraph above, for the probability of choosing surgical therapy in a patient with stage T2, N2a, and M0 laryngeal carcinoma, 74% was entered, while this therapy option would not be chosen with a probability of 26%.

Reviewer 4 Report

see pdf file

Author Response

Comment of reviewer:

English language and style are fine/minor spell check required.

Response to the reviewer:

We conducted a check and correction of spelling and style wherever appropriate.

Comment of reviewer:

  1. Read the text to correct a few typos (e.g. “analyss” line 277).

Response to the reviewer:

Thank you for this valuable remark. We have thoroughly read through the text and made the necessary corrections.

Comment of reviewer:

  1. A directed acyclic graphic could be added, even summarized, to show all the variables or groups of variables (like Figure 3 of Cypko 2019).

Response to the reviewer:

We thank the reviewer for this valuable input. A figure has been added that divides the therapy decision model into its different categories. On the one hand, there is the category TNM stages, which precisely describes the tumor as well as its spread. Secondly, there is the therapy methods category, which includes surgical therapy, radiotherapy, radiochemotherapy, chemotherapy, and immunotherapy. In addition, there is the group Prerequisite for certain therapy options. In our model, this includes the tolerance for chemotherapy. 

Comment of reviewer:

Expert learning and knowledge are not only based on memory (line 34 only mentions the “limited memory of the experts”) but also on other concepts (Polanyi's tacit knowledge for example) or other techniques (“think-silly”, consensus between experts, ...).

Response to the reviewer:

Thank you very much for this remark. Expert knowledge is definitely not the only factor that plays an important role in therapy decisions. As you already mentioned, many factors are involved. Therefore, a therapy decision model should be supportive and in no way substitutive. Among other things, it should support the experts by providing standardized and comprehensive information and therefore help the memorization of the experts. Ultimately, the experts can decide whether to accept the proposal of a therapy decision support system or not. Moreover, they can check whether their intuitive decision is congruent with the therapy model. Especially for physicians who are just starting on their career, a therapy decision system may very well be a very important factor of learning and understanding.

Comment of reviewer:

Line 39, specify that those who advocate “treatment support systems” suggest helping experts and definitely not replacing them.

Response to the reviewer:

Thank you very much for this addition. In any case, treatment decision support systems serve only as an assistant to the experts and cannot and should not replace them in any way. Due to the large amount of data and information flow, it is not easy for experts to consider all the important criteria when making a decision. Therefore, in the future, these support systems should reduce this problem. Nevertheless, the experience and expertise of experts remain the crucial factor in making therapy decisions in everyday clinical practice.

Comment of reviewer:

Review the list in lines 43 and 44. Deep learning is a neural network with at least three layers and on the other hand, neural network and deep learning belong to machine learning.

Response to the reviewer:

Thank you for this remark. We have revised and extended the description of the related technologies.

Round 2

Reviewer 1 Report

Authors tried to answer and did their best to improve the paper but my previous concerns remain. I also recommend authors  to use other metrics for strongly unbalanced datasets as is your case (precision-recall curves, F1, etc.). Otherwise results can be biased , even more when the samples are small or not basic statistical analysis is carried out.

Author Response

Comment of reviewer:

Is the research design appropriate? Must be improved.

Response to the reviewer:

Thank you very much for this review. With our work, we aimed to investigate the process of development of a decision model. Therefore, all information important for this purpose and as far as available was collected. Information regarding the disease, about the patients, expert procedures as well as the available guidelines for therapy decisions were considered. The therapy model was graphically presented in the software GeNIe 2.0 and the probabilities for the different therapy options were used based on the NCCN guidelines for head and neck tumors. For verification and validation, the therapy options suggested by our decision model were compared with the actual therapy performed.

For the therapy options that were clustered in our dataset, the model yielded good results. The presented validation analysis is intended as a proof-of-concept approach to calculate or predict the treatment from clinical primary data. In this proof-of-concept analysis, our model showed good results, and we plan to extend the model in the future, as well as to increase the number of cases in further validation series. So far, we had the presented number of 92 cases in our repository to validate our model due to the single center approach. Still, the accuracy in all different treatment options is very consistent – inspite of the varying number of cases – which suggests that a larger and more balanced data set may support the presented analysis. We aim to conduct further validation to improve the validity of the model, but regard these issues future work that surpasses the scope of the presented work.

We performed additional analyses of the model and added 3 figures and a table to our manuscript. The figures display the Precision-Recall-Curves for the main three therapy options surgery, radiotherapy, and radiochemotherapy. These curves plot the positive predictive value against the true positive rate. Curves for chemo- and immunotherapy were omitted because of missing positive cases in the test data, and therefore graphs would be unusable. All curves show a high sensitivity for one specific option. Figure 6a shows high sensitivity toward larynx surgery (blue curve), Figures 6b and 6c show high sensitivity against the performance of either radio- or radiochemotherapy (orange curves). This means that with a high recall there are few to no false positive predictions. However, the opposite therapy options (orange curve in Figure 6a, blue curves in Figures 6b and 6c, respectively) reach high recall only with lower precision. Our model’s prediction against surgery or for radiochemotherapy, respectively, reaches a precision of roughly 50% with a recall of only 50%. The precision of outcome values for radiotherapy does not exceed 20% precision (see Figure 6c).

We calculated the F1 score as a harmonic mean of precision and recall. Table 4 shows the results of our model. It is also clearly visible how either one therapy option tends to favor only one of its possibilities. Both, Precision-Recall-Curve and F1, display the flaws in the test data, which are unfortunately unbalanced in terms of containing all possible values in a suitable distribution. This implies that statements from our model should be individually assessed in regard to the model’s sensitivity.

Comment of reviewer:

Are the methods adequately described? Can be improved.

Response to the reviewer:

We thank the reviewer for this comment. Our "Methods" section contains all the information relevant to our work, such as the clinical information on laryngeal carcinoma, the basis of our work, and the details regarding our data set. In the different sections, the exact steps of our modeling approach are listed. The exact procedure of data collection, the creation of the model with the different nodes and edges including the graphical representation of the model, more detailed information about the software used, namely GeNIe 2.0, as well as the process of entering the probabilities into this software and the related guidelines are described in detail.

We also provide detailed information on our data set, with sex ratio, frequency in the different age groups, and frequency of the different T, N, and M stages. In the section "Validation Process," we describe the number of patients whose results were actually evaluated, the procedure of importing the patients of our dataset into GeNIe, and how the validation was performed.

We also stated that we abandoned the idea of a statistical analysis of the model because the limited data set would not promise reliable results. The presented work is a proof-of-concept approach, which allowed for the presented mathematical analysis. In the future, however, we plan to test the model prospectively in a separate project, ideally in a controlled clinical trial.

Comment of reviewer:

Are the results clearly presented? Must be improved.

Response to the reviewer:

Thank you very much for this remark. In the chapter "Results", we describe our dataset and our model in tabular and graphical form and show the reader steps of the practical work with the software GeNIe 2.0. We have included figures that explain an isolated node in more detail, show an overall view of the therapy model, represent the probability input to the software, and divide the therapy model into different categories.  Furthermore, we added an additional analysis and supplemented three figures and one table. The figures display the Precision-Recall-Curves for the main three therapy options: surgery, radiotherapy and radiochemotherapy. All curves show a high sensitivity for one specific option. Figure 6a shows high sensitivity toward larynx surgery (blue curve), Figures 6b and 6c show high sensitivity against the performance of either radio- or radiochemotherapy (orange curves) which means that with a high recall there are few to no false positive predictions. As already mentioned above we calculated the F1 score as a harmonic mean of precision and recall. Table 4 shows the results of our model. Both, Precision-Recall-Curve and F1, display the flaws in the test data, which are unbalanced in terms of containing all possible values in a suitable distribution.

Comment of reviewer:

Are the conclusions supported by the results? Must be improved.

Response to the reviewer:

We thank you for this comment. We have now added new content to the Conclusions section of our manuscript. A therapy decision support model serves as an aid for the experts, for example in the tumor board. It can be used as a small feature within a clinical information system in preparation for the tumor board to illuminate possibilities of the therapy approach in advance or it can be used in the Tumor board itself. However, it should not be used as a basis for decision-making. The experience and knowledge of the experts should still be taken into account. It must always be considered in the context of clinical practice.

Our decision model for laryngeal cancer was created by integrating the most relevant parameters for a general treatment decision. The BN model was validated in a retrospective analysis of 92 patients, who were presented on the tumor board of the University Hospital Leipzig from 2017 to 2020. In this analysis, the calculation of the model from primary data taken from the patient record of 92 patients was compared to the treatment decision of the tumor board. Because of conflicting data, information from 5 patient cases was excluded from our analysis so as not to bias the results. The analyses of our model showed an overall accuracy of 91.0%.

Therapy decision support systems can be a real asset in everyday clinical practice. For the development of a decision model, a large balanced data set is indispensable to achieve better results. To prove that the decision model can be used in practice, validation with patient data from clinical practice has to be performed. With a rather unbalanced data set, accurate predictions cannot be made for certain aspects, in our case chemotherapy and immunotherapy. With a larger and more homogeneous data set, better results can be obtained.

Overall, however, the structure of the model and the formalism are very important. Evidence-based causalities were set. The probabilities that were entered into the GeNIe model are based on guidelines that are also used in clinical practice. Extension and maintenance of the model are important and possible at any time. As soon as new knowledge or new findings occur, the model can be adapted to them.

These results will be tested prospectively and we hope that further optimization and validation will lead to a beneficial clinical decision support system that provides transparent and comprehensive assistance in the decision-making process.

Comment of reviewer:

Authors tried to answer and did their best to improve the paper but my previous concerns remain. I also recommend authors to use other metrics for strongly unbalanced datasets as is your case (precision-recall curves, F1, etc.). Otherwise, results can be biased, even more when the samples are small or not basic statistical analysis is carried out.

Response to the reviewer:

We thank the reviewer for this argument. We performed further analysis and included three figures and one table on this in our manuscript. The figures display the Precision-Recall-Curves for the main three therapy options surgery, radiotherapy, and radiochemotherapy. Because of missing values for positive test data in both cases, curves for chemo- and immunotherapy were omitted. As already explained above the Precision-Recall-Curve and F1, display the flaws in the test data, which are unbalanced in terms of containing all possible values in a suitable distribution. This implies that statements from our model should be individually assessed regarding the model’s sensitivity.
